

# Unexpected lower level of oral periodontal pathogens in patients with high numbers of systemic diseases

Michael T. Shen[1], Betti Shahin[2], Zhengjia Chen[3,4] and Guy R. Adami[1,5]

[1] Oral Medicine and Diagnostic Sciences, College of Dentistry, University of Illinois Chicago, Chicago, IL, United States of America
[2] Restorative Dentistry, University of Illinois Chicago, Chicago, IL, United States of America
[3] Division of Epidemiology and Biostatistics, School of Public Health, University of Illinois at Chicago, Chicago, IL, United States of America
[4] Biostatistics Shared Resource Core, University of Illinois Cancer Center, University of Illinois Chicago, Chicago, IL, United States of America
[5] University of Illinois Cancer Center, University of Illinois Chicago, Chicago, IL, United States of America

## ABSTRACT

**Background**. Periodontal disease is associated with systemic conditions such as diabetes, arthritis, and cardiovascular disease, all diseases with large inflammatory components. Some, but not all, reports show periopathogens *Porphyromonas gingivialis* and *Tannerella forsythia* at higher levels orally in people with one of these chronic diseases and in people with more severe cases. These oral pathogens are thought to be positively associated with systemic inflammatory diseases through induction of oral inflammation that works to distort systemic inflammation or by directly inducing inflammation at distal sites in the body. This study aimed to determine if, among patients with severe periodontal disease, those with multi-morbidity (or many chronic diseases) showed higher levels of periodontal pathogens.

**Methods**. A total of 201 adult subjects, including 84 with severe periodontal disease were recruited between 1/2017 and 6/2019 at a city dental clinic. Electronic charts supplied self-reported diseases and conditions which informed a morbidity index based on the number of chronic diseases and conditions present. Salivary composition was determined by 16S rRNA gene sequencing.

**Results**. As expected, patients with severe periodontal disease showed higher levels of periodontal pathogens in their saliva. Also, those with severe periodontal disease showed higher levels of multiple chronic diseases (multimorbidity). An examination of the 84 patients with severe periodontal disease revealed some subjects despite being of advanced age were free or nearly free of systemic disease. Surprisingly, the salivary microbiota of the least healthy of these 84 subjects, defined here as those with maximal multimorbidity, showed significantly lower relative numbers of periodontal pathogens, including *Porphyromonas gingivalis* and *Tannerella Forsythia*, after controlling for active caries, tobacco usage, age, and gender. Analysis of a control group with none to moderate periodontal disease revealed no association of multimorbidity or numbers of medications used and specific oral bacteria, indicating the importance of severe periodontal disease as a variable of interest.

**Conclusion**. The hypothesis that periodontal disease patients with higher levels of multimorbidity would have higher levels of oral periodontal pathogens is false.

Corresponding author
Guy R. Adami, gadami@uic.edu

Multimorbidity is associated with a reduced relative number of periodontal pathogens *Porphyromonas gingivalis* and *Tannerella forsythia*.

# INTRODUCTION

A major part of aging is the accumulation of chronic diseases with the end result being loss of organ and tissue function (*Li et al., 2021*). Multimorbidity, or the presence of multiple chronic conditions, has been shown to have a linear relationship with mortality and loss of fitness (*France et al., 2012*; *Willadsen et al., 2018*). This contributes to the loss of quality of life with aging. Periodontal disease is the inflammatory disease of the specialized tissue that surrounds and anchors teeth in the jaw. Chronic periodontal disease is a classic disease of aging that occurs almost exclusively in adults and is seen to increase in incidence with old age (*Teles et al., 2012*). With chronic periodontal disease, inflammation results in degradation of the supporting tissue of the teeth, which results in excessive bleeding, loss of normal oral soft tissue and finally loss of enamel and dentin of the teeth (*Eke et al., 2015*; *Lang & Bartold, 2018*). Severe chronic periodontal disease is marked by large losses of the supporting structures of the teeth and even some teeth. The severe form of the disease occurs in more than 11% of those over 65 (*Eke et al., 2015*).

Whether in health or disease, there are hundreds of bacterial taxa that live below the gum line between the tooth surfaces and the mucosa. These bacteria are continually shed into the saliva where they can be measured (*Belstrøm, 2020*; *Belstrøm et al., 2014*). As periodontal disease progresses oral health-associated bacteria, such as gram positive bacteria *Actinomyces* spp. and *Streptococcus* spp., diminish and are replaced by anaerobic gram negative bacteria, such as *Porphyromonas gingivialis, Tannarella forsythia, Treponema denticola* and/or *Prevotella intermedia*, at the site of the periodontia (the subgingiva) and in the saliva (*Curtis, Diaz & Van Dyke, 2020*; *Dewhirst et al., 2010*; *Kistler et al., 2013*; *Lee et al., 2021*; *Schincaglia et al., 2017*). These bacteria are believed to promote periodontal disease by producing factors that subvert host immune responses directly and, by promoting expansion of pathological microbes, resulting in a prolonged abnormal inflammatory response which in time helps to destroy the tissue. These changes in both the bacteria population and the body's response to the bacteria *via* the inflammatory system are believed to cause the disease (*Feres et al., 2016*; *Lee et al., 2021*).

Chronic periodontal disease is similar to other chronic inflammation-based diseases in that there are gradual changes in normal tissue and function due to inappropriate inflammation. It is well recognized that having periodontal disease greatly increases the possibility of having another inflammation- and autoimmune-based disease such as type II diabetes mellitus (T2DM), cardiovascular disease, rheumatoid arthritis, systemic lupus erythematous, cancer, and even mental illness (*Falcao & Bullón, 2019*; *Kaur, White & Bartold, 2013*; *Lockhart et al., 2012*; *Nakib et al., 2013*; *Su et al., 2017*; *Wang, Zhang & Wang,*

*2022*; *Zemedikun et al., 2021*). Periodontal pathogens such as *Porphyromonas gingivialis*, and *Tannarella forsythia* have been shown, in some studies, to be elevated in the oral cavity of those with diseases such as T2DM, cardiovascular disease, rheumatoid arthritis, and psoriasis, which can be explained in part by the increased level of periodontal disease seen in these patients (*Choi et al., 2018*; *Fan et al., 2018*; *Kageyama et al., 2019*; *Peters et al., 2017*; *Su et al., 2019*; *Sun et al., 2017*; *Andriankaja et al., 2011*; *Belstrøm, 2020*; *Casarin et al., 2013*; *Da Cruz et al., 2008*; *Ganesan et al., 2017*; *Renvert et al., 2006*; *Rosenbaum & Asquith, 2016*; *Spahr et al., 2006*). There is much evidence in animal models that periodontal disease can exacerbate or initiate systemic diseases (*Lee et al., 2021*; *Peng et al., 2022*). Specifically, addition of *Porphyromonas gingivalis,* or to a much lesser degree, *Tannerella forsythia*, to gums to induce and exacerbate periodontal disease in rodents has been shown to initiate, exacerbate or increase risk factors for rheumatoid arthritis, atherosclerosis, liver disease and Alzheimer's in rodent models predisposed to those diseases (*Cantley et al., 2011*; *Chukkapalli et al., 2015*; *Costa et al., 2021*; *Kuraji et al., 2016*; *Marchesan et al., 2013*; *Nakahara et al., 2018*; *Xiao et al., 2021*; *Zhou et al., 2021*). In humans, treatment of periodontal disease can result in evidence of reduction in severity of T2DM, atherosclerosis and rheumatoid arthritis (*Al-Katma et al., 2007*; *Bokhari et al., 2012*; *D'Aiuto et al., 2018*; *Khare et al., 2016*; *Ogrendik, 2009*; *Tonetti et al., 2007*).

Some proposed explanations for a causative role for periodontal disease in systemic disease (Fig. 1) include periodontal bacteria-based disturbance of the immune system, for example, by providing antigens that lead to autoimmunity (*Thoden van Velzen, Abraham-Inpijn & Moorer, 1984*; *Van Dyke & Van Winkelhoff, 2013*). A second proposed mode of systemic disease induction/exacerbation is through disturbance of the inflammation process, possibly by inducing cytokines that can become systemic. A third proposed mechanism is the release of host cell toxins or host cell function modifiers that work at a distance. It is also known that bacteria normally associated with chronic periodontal disease can be found in disease sites that include arterial walls, the brain, and malignant tumors (*Lee et al., 2021*; *Peng et al., 2022*; *Sobocki et al., 2022*; *Willis & Gabaldón, 2020*). For that reason, it has also been proposed that oral bacteria may travel to distal sites to cause disease, though that is by no means the only way periodontal disease may affect systemic disease.

A major obstacle to healthy aging is the presence of chronic inflammatory disease (*Kandelman, Petersen & Ueda, 2008*). Only about thirty percent of adults make it to age 65 without some chronic disease and most have an inflammatory component (*Directors, Natonal Association of Chronic Disease, 2020*). Those with periodontal disease are much more likely to have these accompanying diseases (*Falcao & Bullón, 2019*). Yet, remarkably, a small fraction of patients with long-standing severe periodontal disease have few or no other major diseases. An analysis of the saliva bacteria of a patient cohort having none to as many as 8 systemic diseases was performed. This was done to address the hypothesis that severe periodontal disease patients avoiding systemic disease have differences in the bacteria associated with their periodontal disease. The salivary profiles of 201 subjects with none, mild, moderate, and severe periodontal disease were tested for their association with
## Periodontal Disease

MOUTH

Periodontal bacteria  →  Immune cells and cytokines  →  Gum and bone damage

REST OF BODY

## Systemic Disease

**Figure 1 Periodontal disease caused by oral bacteria and oral tissue inflammation contribute to systemic disease but by intermediaries that are not known.** Periodontal pathogens or their antigens, inflammatory cells and/or their cytokines, and/or the gum and tooth damage that results may work to increase the incidence of disease elsewhere in the body.

the number of systemic diseases and conditions, revealing significant differences depending on the overall health of the subject.

## MATERIAL & METHODS

A total of 201 individuals participated in this research study. This group was studied as a whole or divided into 84 subjects clinically evaluated as having severe periodontal disease (Stage III and IV), and 117 subjects with no to moderate periodontal disease as control. All patients were drawn from a cohort of 300 patients at an inner-city dental clinic (*Schwartz et al., 2021*). The main focus was on the group with severe periodontal disease. The suitability of the sample size of this group to discern a 2-fold change in relative abundance of a specific taxa between two subgroups with different levels of multimorbidity at a power of 80 and beta of 20 was calculated based on earlier work with saliva samples data and the variability in levels of the 20 most abundant taxa (*Faul et al., 2007*). Detection of a 2-fold difference in relative abundance would require 35 total subjects, while detection of a 50% difference would require 84.

Inclusion criteria were adult, 20 or more teeth, recent dental and periodontal examination. Exclusion criteria were incomplete medical records, implants, acute oral disease or respiratory illness, antibiotic usage in the last month, oral antimicrobial rinse usage in the last 12 h, food or beverage consumption in the last hour. Standard criteria for periodontal disease for Stage III and Stage IV, severe periodontal disease, were interdental clinical attachment loss of > 5 mm, radiographic bone loss to middle third of tooth, probing depths > 6 mm, and tooth loss (*Caton et al., 2018*). Medication usage was gathered from the electronic health records of dental patients at the University of Illinois Chicago Dental clinic. These were assembled by automated linkage to the patient's pharmacy. Only medications taken long term were included. Medical conditions and diseases were drawn

from a comprehensive list of 114 which all clinic patients were routinely queried about on first appointment and then on a yearly basis (Table S1). For patients with more severe illnesses, the medical history was routinely verified with the patient's primary care provider. Additionally, all notes were manually examined to reveal additional chronic diseases that were not reported at the last yearly visit. Institutional Review Board I at the University of Illinois Chicago approved this protocol 2016-0696. All participants were informed of the implications of this study. Verbal and written consent were obtained by all participants prior to sample collection by the clinician.

## Sample collection, DNA extraction, and sequencing

Patient samples were derived from stimulated saliva accumulated from chewing paraffin wax over 5 min and processed as described earlier (*Kawar et al., 2021*). Saliva as collected was stored on ice for up to 2 h prior to centrifugation at 5,500 ×G for 5′, followed by washing with PBS two times. DNA was extracted from frozen bacteria pellets stored at -70 degrees C using the Quick-DNA Fungal/Bacteria Miniprep Kit (Zymo Research, Irvine, CA, USA). The V1–V3 variable region of bacterial 16S rRNA genes was amplified using the primer set 27F/534R followed at the University of Illinois at Chicago Sequencing Core by a second PCR amplification when sample specific barcodes were added followed by cleanup and sequencing as described earlier (*Schwartz et al., 2021*). Sequencing was performed on the Illumina MiSeq using the MiSeq reagent kit V3, 600 cycles (Illumina, Inc, San Diego, CA, USA). Negative controls were samples that started with $H_2O$ instead of saliva DNA. Additional controls were technical replicates from several donors.

For taxa assignment and measurement, forward sequence reads from the FASTQ files were analyzed using the software package QIIME2 (v2022.2) (*Bolyen et al., 2019*). Sequences were trimmed if the average quality was lower than 25. As a result, the read sequences were truncated at 262nt. DADA2-plugin in QIIME2 was used to denoise the sequence and generate feature data and feature tables for the dataset of DNA sequences (*Callahan et al., 2016*). Taxonomy assignment was done by classify-consensus-blast function using the Blast+ consensus taxonomy classifier to determine 98% match identity of the query sequences to the Human Oral Microbiome Database (v15.22) (*Dewhirst et al., 2010*). Of the 201 samples, there were 37,976 reads per sample on average (range 14,589–57,031). There were 7,633,250 reads in total.

## Statistical analysis

Statistical analysis was performed using MaAsLin2 software for multivariable analysis of the sample data. Compound Poisson Linear Models (CPLM) was used for the data analysis, with minimum abundance set to 4, minimum prevalence at 20%, and the FDR (*q*-value) less than 0.1 (*Mallick et al., 2021*). For statistical analysis of the study population demographics and clinical factors, descriptive statistics were used to summarize the characteristics for patients. Categorical variables were summarized using frequencies and percentages. Relationships between categorical variables were tested with the two-tailed Fisher exact test. Continuous variables were presented as mean, standard deviation, or standard error of the mean. Spearman correlation coefficients were estimated to measure

the relationship between two continuous or ordinal variables and tested with Wald's test. To assess the correlations between categorical clinical factors and numerical variables, $t$-test or ANOVA tests were conducted when data follow a normal distribution, otherwise Wilcoxon rank sum test or Kruskal-Wallis test were used instead (*Kirkman, 1996*).

## RESULTS

### Clinical features of the study group

The initial study group consisted of 201 subjects with no periodontal disease all the way to severe periodontal disease, drawn from a cross-sectional study of adults using an inner city dental clinic. A tally of chronic systemic diseases and conditions was calculated for each patient drawn from a list of 114 (Table S1) that informed a morbidity number for each subject. Demographic data of this group after division based on level of periodontal disease showed higher numbers of these diseases with advanced periodontal disease, as expected (Table 1).

### Periodontal pathogens associated with periodontal disease

As a control experiment the initial study group included subjects without periodontal disease and with various stages of periodontal disease. It was not surprising that multiple variable analysis using MaAsLin2 for the fixed variable periodontal disease stage and specific taxa abundance with correction for tobacco usage, gender, age and number of tooth surfaces with active caries, revealed high proportions of periodontal pathogens (*Sobocki et al., 2022*) in the saliva of those with periodontal disease (Table S2) (*Mallick et al., 2021*). Elevated with increased periodontal disease severity were *Porphyromonas gingivalis*, $p < 1.4 \times 10-6$; *Tanerella Forsythia*, $p < 5.7 \times 10-6$; *Treponema denticola*, $p < 3.9 \times 10-5$; along with Fusobacterium.s__sp._HMT_203 $p < 4.810-5$ and _Fusobacterium.s__nucleatum_subsp._vincentii, $p < 0.040$ taxa DNA. The other possible periodontal disease pathogenic taxa, *Prevotella intermedia*, was not observed to be enriched by this measure and *Aggregatibacter actinomycetemcomitans* was not interrogated.

### Advanced periodontal disease subgroup

A subgroup of patients with severe periodontal disease was formed after stratification of the original dataset. Table 2 shows demographic data of this population after subdivision into three columns based on the numbers of diseases or conditions reported for each patient. A fairly large number of patients reported as relatively free of systemic disease or conditions despite having severe periodontal disease. This group with few or no systemic diseases also showed lower usage of medications and was somewhat lower in age (Table 2).

### *Oral bacterial taxa associated with lack of systemic disease and conditions among a subpopulation all with advanced periodontal disease*

Multivariable analysis was performed using MaAsLin2 software, computing the associations between the fixed variable of number of diseases/conditions and specific taxa abundances for the samples, after correcting for possible confounders, age, gender, tobacco usage, and active dental caries (Table 3). The multi-morbidity count for each subject did not include redundancies and short term illnesses such as acute respiratory illnesses (Table S1). At

**Table 1  Demographics of the 201 subjects.**

| | Clinical indices and demographic data of subject groups | | |
|---|---|---|---|
| | **Periodontal status** | | |
| **Variables** | **No to moderate periodontal disease[1] (*n* = 117)** | **Severe periodontal disease[2] (*n* = 84)** | **Probability statistic[3]** |
| Age[4] | 47.4 ± 1.6 | 56.7 ± 1.6 | *p* < 0.041 |
| Gender (F:M) | 83 F(71%):34 M(19%) | 50 F(60%):34 M(40%) | *p* < 0.099 |
| Tobacco usage (Nonsmokers: Smokers) | 105 NS(90%):12 S(10%) | 70 NS(83%):14 S(17%) | *p* < 0.20 |
| Caries[5] | 3.58 ± 0.68 | 5.61 ± 0.78 | *p* < 0.054 |
| Medication count | 2.59 ± 0.33 | 3.39 ± 0.44 | *p* < 0.14 |
| Disease count | 1.86 ± 0.21 | 2.88 ± 0.30 | *p* < 0.041 |

**Notes.**
[1] Up to Stage II.
[2] Stage III to Stage IV.
[3] Differences in the groups with no to mild versus severe periodontal disease were determined using Student t test or Fisher Exact Test as appropriate.
[4] ± values are shown with SEM.
[5] Number of surfaces with caries.

the species level, 13 taxa were observed to show different levels in subjects with severe periodontal disease but low levels of systemic diseases and conditions. Unexpectedly, at the species level, two red complex species, *Porphyromonas gingivalis* and *Tanerella forsythia*, believed to be causative agents of periodontal disease were at significantly higher levels as systemic disease/conditions decreased (*Gambin et al., 2021*). Two *Actinomyces* species associated with good periodontal health, *gerensceriae* and *graevenitzii*, increased in abundance with decreasing levels of disease.

### Bacterial taxa associated with oral microbiome linked systemic diseases

Again, the examination was restricted to a population with chronic severe periodontal disease. It is reasonable to assume that some systemic disease and conditions do not elicit or are not associated with changes in the oral microbiome. For that reason, a new analysis was undertaken that focused only on diseases and conditions that have been reported to be linked to differences in the oral microbiome, as listed in Table 4 (*Aldulaijan et al., 2020*; *Belstrøm et al., 2020*; *Dong et al., 2021*; *Falcao & Bullón, 2019*; *Jia et al., 2018*; *Lee et al., 2021*; *Sobocki et al., 2022*; *Willis & Gabaldón, 2020*). Multivariable analysis to determine the association of the number of these diseases a patient had and the relative levels of salivary taxa revealed 15 significantly associated taxa at FDR < 0.10 (Table 5). There were lower relative numbers of *Actinomyces graevenitzii* and *Actinomyces gerenceseriae* in the saliva of subjects with zero or low numbers of these systemic diseases. At the same time, 15 bacterial taxa were at higher levels in these same subjects with no or low levels of these systemic inflammation-linked diseases, including periodontal pathogens *Tanerella forsythia* and *Porphyromonas gingivalis*.

Shen et al. (2023), *PeerJ*, DOI 10.7717/peerj.15502

Peer**J**

**Table 2  Demographics of the subset of 84 subjects with severe periodontal disease.**

| | Severe periodontitis stages III & IV | | | | |
|---|---|---|---|---|---|
| **Variables** | **0 to 1 Diseases/ Conditions** [1] **(*n* = 33)** | **2 to 3 Diseases/ Conditions (*n* = 26)** | **Probability statistic**[2] | **4+ Diseases/ Conditions (*n* = 25)** | **Probability statistic**[2] |
| Age[3] | $50.02 \pm 2.6$ | $57.62 \pm 2.8$ | $p < 0.054$ | $64.66 \pm 1.9$ | $p < 0.0001$ |
| Gender (F:M) | 15F(45%):18M(55%) | 17F(65%):9M(35%) | $p < 0.19$ | 18F(72%):7M(18%) | $p < 0.062$ |
| Tobacco Users (Nonsmokers: Smokers) | 27 NS(72%):6S(18%) | 22 NS(85%):4 S(15%) | $p < 0.53$ | 21 NS(84%):4 S(16%) | $p < 0.56$ |
| Caries | $6.42 \pm 1.4$ | $4.35 \pm 1.2$ | $p < 0.27$ | $5.84 \pm 1.5$ | $p < 0.78$ |
| Med count | $0.51 \pm 0.203$ | $3.23 \pm 0.57$ | $p < 0.0001$ | $7.36 \pm 0.66$ | $p < 0.0001$ |

**Notes.**

[1] Subjects are grouped according to the number of diseases/medical conditions reported (from the list in Table S1).

[2] Differences versus the group with 0 to 1 conditions were determined using Student *t* test or Fisher Exact Test as appropriate.

[3] $\pm$ values are shown with SEM.

**Table 3** Multivariable analysis to examine salivary taxa associated with multimorbidity among those with severe periodontal disease ($n = 84$).

| Taxa associated with numbers of diseases and conditions[1] of severe periodontitis subjects ($n = 84$) | | | | |
|---|---|---|---|---|
| **Adjusted for caries, gender, age, tobacco use** | | | | |
| **Taxa** | **Coef[2]** | **Std. Dev.** | ***p*-value** | ***q*-value** |
| Leptotrichia.s__sp._HMT_221 | −1.1639 | 0.2510 | 0.0000 | 0.0020 |
| Tannerella.s__forsythia | −0.9307 | 0.2304 | 0.0001 | 0.0058 |
| Desulfobulbus.s__sp._HMT_041 | −1.6614 | 0.4800 | 0.0005 | 0.0279 |
| Actinomyces.s__sp._HMT_171 | −1.1273 | 0.3426 | 0.0010 | 0.0407 |
| Fusobacterium.s__sp._HMT_203 | −1.6482 | 0.5053 | 0.0011 | 0.0432 |
| Streptococcus.s__australis | 0.5120 | 0.1705 | 0.0027 | 0.0685 |
| Gemella.s__morbillorum | −0.4752 | 0.1565 | 0.0024 | 0.0685 |
| g__Mogibacterium.__ | 0.3913 | 0.1316 | 0.0029 | 0.0730 |
| Porphyromonas.s__gingivalis | −0.8105 | 0.2741 | 0.0031 | 0.0738 |
| Leptotrichia.s__sp._HMT_212 | −0.5460 | 0.1873 | 0.0035 | 0.0793 |
| Capnocytophaga.s__granulosa | −0.6625 | 0.2338 | 0.0046 | 0.0907 |
| Fretibacterium.s__fastidiosum | −0.7203 | 0.2558 | 0.0049 | 0.0911 |
| Streptococcaceae_.XI..G.4..__ | −1.0637 | 0.3839 | 0.0056 | 0.0958 |

**Notes.**
[1] Nonrendundant diseases and conditions for each subject (Table S1).
[2] Negative coefficient indicates taxa that decrease with high numbers of disease/conditions.
[3] Taxa with differential level of significance at False Discovery Rate of $q < 0.1$.

### Control experiment to determine association of multiple medication usage with oral bacteria in a population with mild or no periodontal disease

It is possible that the association of bacteria taxa and the degree of multimorbidity observed above is due to the parallel increase in medication number that occurs. Due to the strong dependence of medication number on numbers of systemic diseases (Table 2), controlling for medication usage in the above multiple variable analysis resulted in no significant associations seen between disease number and oral taxa (data not shown). For that reason, a subset of subjects from the original population was examined. It consisted of 117 adults with none, mild, and moderate periodontal disease (Table S3). This group showed a similar range in number of diseases/conditions, level of medication usage, and the five most popular medications were identical to that of the groups with severe periodontal disease, HMG-CoA reductase inhibitors (atorvastatin), selective serotonin uptake inhibitors (citalopram), Angiotensin-converting enzyme inhibitors (lipsinopril), metformin, and aspirin. As might be expected this population was younger and showed lower numbers of diseases and conditions per subject (Table 1). A tally was made of the numbers of medications prescribed to each subject. Multivariable analysis was done using MaAsLin2 to compute the associations between the fixed variable, medication count, and the relative taxa abundancies among the samples, after adjusting for possible confounders, age, tobacco usage, gender and caries. One taxon was found to be directly associated with decreased medication number, Saccharibacteria_.TM7._.G.3. s__bacterium_HMT_351. Curiously, a similar analysis of bacteria taxa associated with disease/condition number

**Table 4  Systemic diseases/conditions reported linked to distinct oral microbiome.**

Alzheimer's Disorder

Barrett's Esophagus

Cancer

Cardiovascular Disease

Celiac Disease

Cervical myelodysplasia

Chronic obstructive pulmonary disease

Congestive Heart Failure

Dementia

Depression

Emphysema

Epilepsy

Gastroesophageal Reflux Disease

Glaucoma

Gout

Hypercholesterolemia

Hyperlipidemia

Hypertension

Idiopathic Bowel Disease

Kidney Disease

Multiple Sclerosis

Muscle Weakness

Neuropsychiatric Disorder

Obesity

Osteoarthritis

Peripheral Vascular Disease

Polycystic Kidney Disease

Post Traumatic Stress Disorder

Psoriasis

Pulmonary Hypertension

Rheumatoid Arthritis

Seizures

Sjogren's Syndrome

Stroke

Systemic Lupus Erythematosus

T1DM

T2DM

**Notes.**

[1] List of condition and disease (Table S3) is limited to those known to be associated with oral microbial taxa differences.

[2] Negative coefficient indicates taxa that decrease with high number of disease/conditions.

[3] Taxa with differential level of significance at False Discovery Rate of $q < 0.1$.

**Table 5  Multivariable analysis to examine salivary taxa associated with high numbers of diseases and conditions previously linked to distinct oral microbiome profiles, among those with severe periodontal disease ($n = 84$).**

| | Taxa associated with numbers of diseases and conditions subjects with severe periodontitis ($n = 84$) | | | |
|---|---|---|---|---|
| | Adjusted for caries, gender, age, tobacco use | | | |
| Taxa | Coef[1] | Std. Dev. | p-value | q-value |
| Leptotrichia.s__sp._HMT_221 | −0.9844 | 0.2444 | 0.0001 | 0.0060 |
| Gemella.s__morbillorum | −0.5162 | 0.1484 | 0.0005 | 0.0260 |
| Porphyromonas.s__gingivalis | −1.0355 | 0.3103 | 0.0008 | 0.0345 |
| Tannerella.s__forsythia | −0.6964 | 0.2083 | 0.0008 | 0.0345 |
| Corynebacterium.s__durum | −0.5959 | 0.1864 | 0.0014 | 0.0438 |
| Streptococcus.s__gordonii | −0.4589 | 0.1440 | 0.0014 | 0.0442 |
| g__Capnocytophaga.__ | −0.4594 | 0.1446 | 0.0015 | 0.0442 |
| Actinomyces.s__sp._HMT_171 | −0.9474 | 0.3007 | 0.0016 | 0.0442 |
| Capnocytophaga.s__granulosa | −0.6892 | 0.2189 | 0.0016 | 0.0442 |
| Leptotrichia.s__sp._HMT_212 | −0.5345 | 0.1757 | 0.0024 | 0.0559 |
| Corynebacterium.s__matruchotii | −0.4558 | 0.1516 | 0.0026 | 0.0602 |
| Actinomyces.s__gerencseriae | 0.5750 | 0.1966 | 0.0034 | 0.0702 |
| Desulfobulbus.s__sp._HMT_041 | −1.5809 | 0.5538 | 0.0043 | 0.0819 |
| Actinomyces.s__graevenitzii | 0.3961 | 0.1400 | 0.0047 | 0.0863 |
| g__Rothia.__ | −0.5373 | 0.1945 | 0.0057 | 0.0953 |

**Notes.**

[1] List of condition and disease (Table S3) is limited to those known to be associated with oral microbial taxa differences.

[2] Negative coefficient indicates taxa that decrease with high number of disease/conditions.

[3] Taxa with differential level of significance at False Discovery Rate of $q < 0.1$.

among the no to moderate periodontal disease groups revealed no taxa differences (data not shown).

## DISCUSSION

In that periodontal disease and periodontal bacteria have been associated with an ever-increasing number of systemic diseases, it is remarkable that some people of advanced age with chronic severe periodontal disease are systemically healthy (*Falcao & Bullón, 2019*; *Willis & Gabaldón, 2020*). A simple hypothesis was that the bacteria, such as *Porphyromonas gingivalis,* associated with systemic diseases such as cancer, cardiovascular disease, autoimmune disease, would be at relatively lower levels in the patients who lacked these and other chronic diseases compared to the patients who had multimorbidity. Given the notion that almost all chronic diseases are largely driven at least in part by inflammatory immune cells, and periodontal disease alters and increases chronic immune cells (*Hajishengallis, 2022*), no effort was made to select diseases to count toward the multimorbidity index. It was thought that this would avoid bias, though obvious acute disease like short term respiratory infections were not included. Instead, a simple tally of chronic diseases and several conditions was made from a list of over 114 entries (Table S1). This was studied in a group of 84 subjects with severe periodontal disease. This first

analysis, providing patients of severe periodontal disease with a "morbidity number" based on the number of chronic systemic diseases and conditions suggested that, paradoxically, individuals with none or lower numbers showed orally higher relative levels of periodontal pathogens, including *Porphyromonas gingivalis* and *Tannerella forsythia*, when compared to those with multiple morbidity. Correlation analyses showed that the relationship between levels of *Porphyromonas gingivalis* and multimorbidity number, for each subject, was measured as a Spearman correlation coefficient of $-0.252$ with $p < 0.021$ from Wald's test. Similarly, the Spearman correlation coefficient for *Tannerella forsythia* and multimorbidity number was $-0.262$ with $p < 0.016$. The directions of measured correlations agree with the corresponding ones from the multiple variable analysis for both relationships, suggesting the validity and robustness of using a Linear Model in the analyses for their relationships.

A similar analysis was done among the same set of patients focusing on systemic diseases reported to have a unique oral microbiome different than that seen in a healthy population, which includes arthritis, T2DM, autoimmune disease, hypertension, cancer, diseases long suspected to be associated with periodontal disease and its pathogens (Table 4) (*Aldulaijan et al., 2020*; *Belstrøm et al., 2020*; *Dong et al., 2021*; *Falcao & Bullón, 2019*; *Jia et al., 2018*; *Lee et al., 2021*; *Sobocki et al., 2022*; *Willis & Gabaldón, 2020*). The rationale was to increase sensitivity to differences in oral bacteria by narrowing the list of diseases counted to those known to be associated with taxa differences in the oral microbiome. From the multivariable analysis, more significant associations were determined after focusing on this subset of oral microbiome relevant systemic diseases, with 13 taxa being of lower abundance with higher amounts of disease, and two taxa being of higher abundance with higher amounts of disease (FDR < 0.1, correction for tobacco usage, active caries, gender age). We again found lower abundances of red complex periopathogens and other periodontal disease-associated bacteria in the saliva of subjects with increasing amounts of systemic disease (Table 5) (*Gambin et al., 2021*).

One possible explanation for the negative association of numbers of diseases and periodontal pathogens was that patients with more systemic disease are known to use more prescription medications (*Payne et al., 2014*). Maybe some of these drugs are directly toxic to periopathogens. There is ample evidence that a number of medications, including proton pump inhibitors (PPIs), metformin, nonsteroidal anti-inflammatory drugs (NSAIDs), opioids, statins, and antipsychotics have the potential to alter gut and oral bacteria levels by direct effects and indirect effects on the bacteria or the bacteria environment (*Falcao & Bullón, 2019*; *Falony et al., 2016*; *Le Bastard et al., 2018*). Testing for an association between number of medications used and salivary bacterial taxa in a population with none to moderate periodontal disease revealed no significant associations. This result suggests that direct stimulatory or inhibitory effects of medications on microbes increasing with increased number of medications is unlikely. However, that does not rule out medication number having an effect on bacteria in patients with severe periodontal disease.

One source of error is the possibility of higher levels of periodontal treatment, and thus lower levels of periodontal bacteria in subjects with multiple diseases. However, a careful examination of treatment records among patients with severe periodontal disease showed that those with one or fewer chronic systemic diseases and those with greater

amounts of disease had a similar spectrum of times since their last periodontal treatment relative to the date of sample collection. Notably, most patients had either not been treated for periodontal disease in a year or more prior, or had never received any form of periodontal therapy at all. Important confounders of saliva microbiome diversity, active caries, periodontal disease and antibiotic usage had been adjusted and controlled for, and the other contributors to taxa diversity, such as tobacco usage (estimated at 3% of the variability of the saliva microbiome) were also considered (*Belstrøm, 2020*; *Poulsen et al., 2022*; *Stewart et al., 2018*). Diet, which is thought to contribute though to a lesser degree to the oral microbiome makeup, was not monitored (*Wade, 2021*; *Wells et al., 2022*) (*Sekundo et al., 2022*). Also, no account of severity of systemic disease was made though all except one subject was ambulatory, and all were free of acute respiratory illness.

It is possible that the lower levels of oral periodontal pathogens with increased levels of systemic disease may be due to changes in the inflammatory or immune system that are more likely to occur when there are multiple systemic diseases. This may be analogous to the change in immunity that has been proposed to happen as pancreatic cancer progresses (*Willis & Gabaldón, 2020*). For example, perhaps the body's immune response to oral pathogens is more efficient in the presence of these diseases. The reason is not known but one speculative model would be that in some cases the onset of systemic disease is associated with translocation of oral bacteria or fragments to the blood (*Aarabi et al., 2015*; *Sobocki et al., 2022*). This simultaneously would increase opportunities for induction of an immune response and the production of antibody against the translocated bacteria. As a result, these targeted bacteria would then decrease at all body sites including the oral cavity. There is some evidence for increased levels of antibodies against periodontal pathogens, such as *Porphyromonas gingivalis* and *Tannerella Forsythia,* in the blood in the presence of specific systemic diseases (*Aoki et al., 2020*; *Colhoun et al., 2008*; *Franciotti et al., 2021*; *Hanaoka et al., 2013*; *Jaramillo et al., 2013*; *Okada et al., 2011*) though it is not always observed (*Lund Håheim et al., 2022*; *Qi et al., 2020*). With multiple illnesses associated with oral bacteria, the probability of this change in immune response to oral bacteria might increase. One candidate target would be *Porphyromonas gingivalis* as it has been associated with multiple chronic systemic diseases (*Lee et al., 2021*; *Willis & Gabaldón, 2020*). This model would suggest that for periodontal pathogens in the setting of established chronic periodontal disease and systemic disease, there is a balance between a push for high perio-pathogen bacteria due to periodontal disease and low numbers due to differences in systemic inflammation and immunity that occur with multiple chronic illnesses. This is supported by the observations that elevated levels of oral *Porphyromonas gingivalis* and other periodontal pathogens tied to systemic disease are clearly seen when the oral samples are taken prospectively prior to the disease diagnosis, be it cancer or cardiovascular disease (*Choi et al., 2018*; *Fan et al., 2018*; *Peters et al., 2017*; *Sun et al., 2017*). When oral bacteria are examined in patients with long-term chronic disease, such as rheumatoid arthritis, periodontal disease-associated bacteria often are not elevated (*Zhang et al., 2015*). Notably in the case of T2DM there are two reports of lower levels of periodontal disease pathogen *Tanerella forsythia* and one for *Porphyromonas gingivalis* when compared to controls without T2DM (*Casarin et al., 2013*; *De la Cruz Pena et al., 2018*). One might speculate

with multiple diseases in a single patient, the negative effects on periodontal pathogens are additive, resulting in obvious low levels of these microbes as seen in this study.

## CONCLUSION

As expected, the saliva of patients with periodontal disease show elevated levels of periodontal pathogens such as *Porphyromdonas gingivalis*, and *Tanerella forsythia* that were higher with increased severity of periodontal disease (Table S2). Unexpectedly, among patients with severe periodontal disease, the main difference in saliva bacteria between those with multiple systemic diseases and the healthier group with fewer systemic diseases is that the healthy group had higher proportions of periodontal pathogenic bacteria (Tables 3, 5). One speculation is that patients with severe periodontal disease but systemically healthy, will have differences in their inflammation or immunity state that allows them to sustain high levels of periodontal pathogens in the mouth but minimal chronic disease elsewhere in the body. Finding the explanation for this phenomenon may reveal better how periodontal disease and many systemic diseases are related.

### Funding

This work was funded by the Department of Oral Medicine and Diagnostic Sciences University of Illinois Chicago, College of Dentistry. The Research Open Access Publishing (ROAAP) Fund of the University of Illinois Chicago provided financial support towards the open access publishing fee for this article. The research reported in this publication was supported in part by the University of Illinois Cancer Center Biostatistics Shared Resource (BSR). The funders had no role in study design, data collection and analysis, decision to publish, or preparation of the manuscript.

### Grant Disclosures

The following grant information was disclosed by the authors:
The Department of Oral Medicine and Diagnostic Sciences University of Illinois Chicago, College of Dentistry.
Research Open Access Publishing (ROAAP) fund of the University of Chicago.
The University of Illinois Cancer Center Biostatistics Shared Resource (BSR).

### Competing Interests

The authors declare there are no competing interests.

### Author Contributions

- Michael T. Shen conceived and designed the experiments, performed the experiments, analyzed the data, prepared figures and/or tables, authored or reviewed drafts of the article, and approved the final draft.
- Betti Shahin conceived and designed the experiments, performed the experiments, authored or reviewed drafts of the article, and approved the final draft.
- Zhengjia Chen analyzed the data, authored or reviewed drafts of the article, and approved the final draft.
- Guy R. Adami conceived and designed the experiments, performed the experiments, analyzed the data, prepared figures and/or tables, authored or reviewed drafts of the article, and approved the final draft.

## Human Ethics

The following information was supplied relating to ethical approvals (i.e., approving body and any reference numbers):

University of Illinois Institutional Review Board 1 granted Ethical approval to do this study.

## Data Availability

The data is available at National Center for Biotechnology Information Sequence Read Archive: PRJNA739492 and PRJNA674379.

## Supplemental Information

Supplemental information for this article can be found online at http://dx.doi.org/10.7717/peerj.15502#supplemental-information.

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
