# Peer review of "Unexpected lower level of oral periodontal pathogens in patients with high numbers of systemic diseases"

_PeerJ, doi:10.7717/peerj.15502_

## Round 0.1 · original submission · Minor Revisions

Based on the Reviewers' Comments, minor revision is needed.

Reviewer 1 ·

Basic reporting

In the abstract, the background is too long. It is better to reduce the contents in the background.
The authors mention the results of animal studies. Are there human studies? It is better to add.
The Result section is misleading and it should be the method section. Please correct and add the Results section. Please present the conclusion in a better way.
In the introduction, it is informative to add more on chronic diseases with periodontitis. The authors can add a Table or Figure for better explanation and understanding.

Discussion
More discussion is needed with more explanation regarding the statement “…the healthy group
375 had more periodontal pathogenic bacteria”. Recent literature needs to be added to support this statement.

Experimental design

Method
As periodontal disease is a multifactorial pathogen origin, how the two bacteria were selected? Need more explanation.
Details on the sample selection isis missing. Please add the details on the sample size selection and calculation.
It is better to add the timeline of this study. From which year to which year was done?
Add details on the statistical analysis software package QIIME2 e.g. company, country, etc. This should be followed for all the instruments used in this research.

Validity of the findings

Results.
It will be useful to compare the results between healthy and periodontitis.

Additional comments

My decision is to reconsider after the revision.

·

Basic reporting

This article is clear on its content and is unambiguous and professional. The literature search is adequately done and well explained.

Experimental design

Sounds good

Validity of the findings

ok

Additional comments

Abstract
The Objective and Method sections is missing. Please add those.

Method
Add details on the sample selection calculation.
The authors studied on the two bacteriaPorphyromonasgingivialisand Tannerella
Forsythia. Why these two bacteria are selected give more explanation.

Result
Comparison between healthy and periodontitis is missing.

Discussion
Please add more recent literature and discuss the results

Conclusion
It is too long. Make it more concise.

Reviewer 3 ·

Basic reporting

Dear authors,
I would like to thank you a lot for this amazing work. I have a few comments for your article please.

First, the abstract would be more clear if it includes the study sample and how it was managed (divided into etc..).

Secondly, please define clearly the multimorbidity in your introduction, and try to be consistent on your writing, either you use the term multimorbidity or multiple chronic conditions for the entire paper.

Thirdly, you have mention arthritis, T2DM, autoimmune disease, hypertension and cancer, why have you mention only these conditions, Are these the only conditions that were related to oral pathogens? please clarify?
furthermore, have you included certain diseases in your analysis, what are they? are they the aforementioned disease?

please indicate the level of significance on the tables footnotes, P<0.001, P<0.01 and P<0.05

Experimental design

No comment

Validity of the findings

No comment

·

Basic reporting

The article was well written. However I did have a few points.

63 - I felt there was a word missing. I would add (being) after end result in the sentence (end result loss of organ)

121 - The way of referencing the CDC by adding the link should be in the references not the article

125 - Long sentence. Either divide them into two or shorten it

166 - (ribosomal RNA rRNA) was written twice in different forms with no brackets.

277 - The first word of the discussion can be changed instead of (in that)

Experimental design

No comment

Validity of the findings

No comment

---

## Round 0.2 · accepted · Accept

I have read the reviewers' comments and the manuscript's recent version. According to the comments, I can confirm that the authors have made substantial changes to the manuscript.